# Prostate Cancers Invisible on Multiparametric MRI: Pathologic Features in Correlation with Whole-Mount Prostatectomy

**DOI:** 10.3390/cancers15245825

**Published:** 2023-12-13

**Authors:** Aritrick Chatterjee, Alexander Gallan, Xiaobing Fan, Milica Medved, Pranadeep Akurati, Roger M. Bourne, Tatjana Antic, Gregory S. Karczmar, Aytekin Oto

**Affiliations:** 1Department of Radiology, University of Chicago, Chicago, IL 60637, USA; xfan@uchicago.edu (X.F.); mmedved@uchicago.edu (M.M.); gskarczm@uchicago.edu (G.S.K.); aoto@uchicagomedicine.org (A.O.); 2Sanford J. Grossman Center of Excellence in Prostate Imaging and Image Guided Therapy, University of Chicago, Chicago, IL 60637, USA; 3Department of Pathology, Medical College of Wisconsin, Milwaukee, WI 53226, USA; agallan@mcw.edu; 4Department of Biology, Loyola University, Chicago, IL 60611, USA; 5Discipline of Medical Imaging Science, Sydney School of Health Sciences, Faculty of Medicine and Health, The University of Sydney, Sydney, NSW 2006, Australia; roger.bourne@sydney.edu.au; 6Department of Pathology, University of Chicago, Chicago, IL 60637, USA; tatjana.antic@uchospitals.edu

**Keywords:** prostate cancer, MRI, invisible lesion, tissue composition

## Abstract

**Simple Summary:**

We investigated why some prostate cancers (PCas) are not identified on multiparametric MRI (mpMRI) by using ground truth reference from whole-mount prostatectomy specimens. A total of 61 patients with PCa underwent 3T mpMRI followed by prostatectomy. Lesions visible on MRI prospectively or retrospectively identified after correlating with histology were considered “identified cancers” (ICs). Lesions that could not be identified on mpMRI were considered “unidentified cancers” (UCs). Pathologists marked the Gleason score, stage, size, and density of cancer glands and performed quantitative histology to calculate the tissue composition. The UCs were significantly smaller, had lower Gleason scores and clinical stage lesions, with a lower density of cancer glands, compared with the ICs. Independent from size and Gleason score, tissue composition differences, specifically, the higher lumen and lower epithelium in UCs (associated with higher T2 and ADC), can explain why some of the prostate cancers cannot be identified on mpMRI.

**Abstract:**

We investigated why some prostate cancers (PCas) are not identified on multiparametric MRI (mpMRI) by using ground truth reference from whole-mount prostatectomy specimens. A total of 61 patients with biopsy-confirmed PCa underwent 3T mpMRI followed by prostatectomy. Lesions visible on MRI prospectively or retrospectively identified after correlating with histology were considered “identified cancers” (ICs). Lesions that could not be identified on mpMRI were considered “unidentified cancers” (UCs). Pathologists marked the Gleason score, stage, size, and density of the cancer glands and performed quantitative histology to calculate the tissue composition. Out of 115 cancers, 19 were unidentified on MRI. The UCs were significantly smaller and had lower Gleason scores and clinical stage lesions compared with the ICs. The UCs had significantly (*p* < 0.05) higher ADC (1.34 ± 0.38 vs. 1.02 ± 0.30 μm^2^/ms) and T2 (117.0 ± 31.1 vs. 97.1 ± 25.1 ms) compared with the ICs. The density of the cancer glands was significantly (*p* = 0.04) lower in the UCs. The percentage of the Gleason 4 component in Gleason 3 + 4 lesions was nominally (*p* = 0.15) higher in the ICs (20 ± 12%) compared with the UCs (15 ± 8%). The UCs had a significantly lower epithelium (32.9 ± 21.5 vs. 47.6 ± 13.1%, *p* = 0.034) and higher lumen volume (20.4 ± 10.0 vs. 13.3 ± 4.1%, *p* = 0.021) compared with the ICs. Independent from size and Gleason score, the tissue composition differences, specifically, the higher lumen and lower epithelium in UCs, can explain why some of the prostate cancers cannot be identified on mpMRI.

## 1. Introduction

Prostate cancer (PCa) is the most common non-cutaneous cancer in men in the United States, with one out of nine men affected by it [1]. Magnetic resonance imaging (MRI) is increasingly being used for prostate cancer diagnosis and for guiding biopsies due to its advantages over traditional prostate-specific antigen (PSA) screening and transrectal ultrasound (TRUS)-guided biopsies in its ability to reliably visualize the whole prostate noninvasively [2,3,4]. MRI not only provides information about the presence of PCa, but also about the location and size of PCa lesions [5,6]. Despite MRI’s good soft tissue contrast, higher sensitivity, and negative predictive value for PCa detection, a large number of prostate cancers still go undetected [3,7,8,9]. It has been noted that up to 30% of clinically significant cancers can be missed even by expert radiologists on multiparametric MRI (mpMRI) using the PIRADS guidelines [9,10,11,12], with a large variation seen between radiologists and imaging centers [13,14].

An improved understanding of why some lesions are missed by mpMRI may be critical to improving prostate MR imaging protocol and may lead to increased diagnostic accuracy. The few studies that have looked into the characteristics of these missed cancers suggest that missed cancers tend to be of smaller size or volume, have a lower Gleason score and PSA density, and are located in a specific location (apical or anterior) compared with cancer lesions that were identified on mpMRI [9,10,12,15,16,17,18]. However, only a handful of studies have investigated histologic differences between identified and unidentified cancer lesions [19,20]. A significant fraction of missed cancers cannot be explained by their small size or low Gleason score, and some biophysical explanation as to why some cancers are missed is needed. In addition, mpMRI is also known to underestimate the true index lesion pathological volume [21,22] and represents the foundation to further investigate invisible vs. visible tumors. Because the contrast on prostate MRI scans heavily depends on the tissue’s microstructures and their distinct MR properties [23], we hypothesize that histological features (such as tissue composition and cancer cell density) may impact the MR visibility of certain cancers. Therefore, this study aims to investigate why some prostate cancers are not identified on mpMRI using ground truth reference from whole-mount prostatectomy specimens.

## 2. Materials and Methods

### 2.1. Study Patients

This institutional-review-board-approved study involved a retrospective analysis of prospectively acquired data. It was conducted with prior informed patient consent and was HIPAA compliant. Inclusion criteria for this study included patients with prior biopsy-proven prostate cancer that underwent prostate mpMRI on a 3T scanner, followed by subsequent radical prostatectomy with whole-mount processing and digitization of histology slides. Exclusion criteria included the receipt of radiation or hormonal therapy prior to MRI that could have led to alterations in prostatic signal on MRI and impaired renal function (glomerular filtration rate or GFR less than 60 mL/min). Of the 108 patients that had initially consented and were imaged at our research center between March 2014 and June 2017, only 61 patients that fit the criteria were recruited for this study. There is an overlap between some of the research subjects used in this study and some of our previously published works [24,25], but none of the previous papers investigated why some prostate cancers are not identified on mpMRI.

### 2.2. MR Imaging

Patients underwent preoperative multiparametric MRI using a 3T Philips Achieva MR scanner along with an endorectal coil (Medrad, Warrendale, PA, USA) and a 16-channel phased array coil (Philips, Eindhoven, Netherlands) placed around the pelvis. The prostate mpMRI protocol included axial and coronal T2-weighted, axial multi-echo T2-weighted, axial diffusion-weighted, and axial dynamic contrast-enhanced images. Typical MR imaging parameters that were used are described in further detail in Table 1. 

### 2.3. Histology Processing

The subjects underwent radical prostatectomy. The excised prostate specimens were fixed overnight in 10% buffered formalin. The fixed organ was then serially sectioned transversely (4 mm slice thickness), approximately in the same plane as the axial MR images. Sectioned tissue was embedded in paraffin and microtomed into 5 μm sections, and whole-mount hematoxylin and eosin (H&E) stained slides were made. The slides were evaluated for prostate cancer by expert pathologists (T.A. and A.G., 15 and 5 years’ experience, respectively) and all PCas were marked on the histologic slides for correlation with MR images. However, only cancer lesions meeting the minimum size criteria of 5 mm × 5 mm were included in the analysis. The whole-mount sections were scanned and digitized using a brightfield Olympus VS120 whole-mount digital microscope (Olympus Corporation, Waltham, MA, USA) at 20× magnification. 

### 2.4. MR Image Analysis

MRI and histology sections were matched by the consensus of an experienced radiologist (18 years’ experience with prostate MRI), pathologist (15 years’ experience), and medical physicist (8 years’ experience). The lesions that were visible on conventional multiparametric MRI (T2-weighted, diffusion-weighted images along with apparent diffusion coefficient maps and dynamic contrast-enhanced images) and were prospectively (blinded to pathology results) or retrospectively identified after correlating with histology were considered “identified cancers”. Lesions that could not be identified (invisible) on mpMRI even after radiological–pathological correlation were considered “unidentified cancers”. 

Using the axial T2-weighted images as the reference image, other mpMRI sequences—diffusion-weighted and contrast-enhanced MRI—were co-registered using rigid registration in 3D Slicer (https://github.com/rcc-uchicago/PCampReview, accessed on 1 November 2016) and matched with corresponding histological sections. Subsequently, MR images were analyzed by an expert radiologist (A.O., 15 years’ experience with prostate MRI), and regions of interest (ROIs) were marked on axial T2-weighted images on regions of known PCa verified on whole prostatectomy sections. The ROIs for the cancer lesions were drawn by the radiologist based on outlines by pathologists on whole-mount histologic sections. These ROIs drawn on T2W images were then transferred to other mpMRI sequences, keeping the same shape and size as on the T2W images using 3D Slicer, similar to previous studies [24,25,26]. 

Quantitative analysis of MRI data was performed using in-house programs in MATLAB v2019a (Mathworks, Natick, MA, USA) and Interactive Data Language (IDL, Harris Geospatial Solutions, Boulder, CO, USA). Apparent diffusion coefficient (ADC) and quantitative T2 maps were calculated from diffusion-weighted and multi-echo T2-weighted images, respectively, using a mono-exponential signal decay model using custom written code in MATLAB on a voxel-by-voxel basis, and mean values (ADC and T2) for the ROIs were calculated. For DCE-MRI, the percent signal enhancement (PSE) curve (relative to unenhanced image intensity) is calculated using the formula
$$PSE(t)=\frac{{S(t)-S}_{0}}{S_{0}}\times 100$$
where S_0_ is the baseline signal intensity and S(t) is the DCE signal at each time point (t). Subsequently, PSE vs. time curves were analyzed in IDL using the empirical mathematical model (EMM) described in Fan et al. [27] with the following equation:$$PSE(t)=A(1-e^{-\alpha t})e^{-\beta t}$$
where A (%) is the amplitude of the relative percent signal enhancement curve or PSE, α describes the signal enhancement due to contrast agent uptake, and β is the washout rate.

### 2.5. Pathology Image Analysis

Two experienced, board-certified genitourinary pathologists (T.A. and A.G., with 15 and 5 years of experience with prostate pathology, respectively), working in consensus, marked the Gleason score, pathologic stage, and size (dimensions at the section of largest extent) for all cancers. 

We further investigated all cancers that were undetected, along with identified cancers with matching size and grade (Gleason 3 + 3 and 3 + 4). The relative density, which is defined as the density of cancer glands with respect to surrounding tissue (category or scale 1–5, with 5 being highly dense compared with surrounding tissue), and the absolute density scale, which is defined as how sparse or dense the cancer glands are in the lesion [28] (category 1–4, with 4 being highly dense cancer), were determined visually by the experienced pathologists. In cancers with Gleason score 3 + 4, the percentage of Gleason pattern 4 was also determined. 

The cancer regions of interest (rectangular area) were extracted from digitized whole-mount images using the BIOP VSI reader plugin on ImageJ v1.51(National Institutes of Health, Bethesda, MD, USA). The software was used to crop the intended area and convert the digitized pathology slides from the proprietary virtual slide image (.vsi) file format (Olympus proprietary format) to the commonly used tagged image file format (TIFF). To avoid errors during tissue segmentation, luminal secretions were removed from the glandular space using Microsoft Paint (https://mspaint.humanhead.com/, accessed on 1 November 2016) (Microsoft Corporation, Redmond, WA, USA). Quantitative histology was then performed to calculate fractional volumes of tissue components using image segmentation of H&E-stained prostate tissue to separate the stroma, epithelium, and lumen using Image Pro Premier v9.2. This was performed on the basis of color, intensity, morphology, and background using the “Smart Segment” functionality, which was described in a previous study [23]. Like in the previous study, segmentation performed with the software was visually inspected by the consensus of a pathologist and medical physicist and rectified iteratively until the final segmented image was determined to have fewer than 5% errors. The tissue fractional volumes were calculated using the “Count” functionality and defined as the percentage of image pixels that were segmented as that tissue type. The segmentation function of this tool has been validated to estimate prostate tissue composition and used in previous studies with reported correlation greater than 0.9 with manual segmentation [23,29,30].

### 2.6. Statistical Analysis

Statistical analysis was performed using SPSS v28 (IBM Corporation, Armonk, NY, USA). A one-sided *t*-test (parametric test for scalar data: lesion size, ADC, T2, DCE parameters, proportion of Gleason 4 pattern, and tissue volumes) or Mann–Whitney U test (nonparametric test for categorical/ordinal data: Gleason score, stage, and relative and absolute density of lesions) was performed to determine whether any significant differences between the identified and unidentified cancers were present. The significance level was set as *p* < 0.05.

## 3. Results

### 3.1. Participant and Tumor Characteristics

A total of 61 patients who fit the criteria (imaged at our research center, followed by subsequent prostatectomy and digitized pathology) were recruited for this study. The mean age (± standard deviation) of all included men was 59 ± 8 years (range 40–76 years), and mean PSA was 8.2 ± 7.9 ng/mL (range 0.8–66.1 ng/mL). A total of 115 confirmed cancers met the minimum size criteria and were included in the study. 

Only 19 of 115 (17%) confirmed cancerous lesions were unidentified on mpMRI. These unidentified cancers came from 17 subjects, while the identified lesions came from 57 subjects. Of the 96 identified lesions, 83 lesions were found prospectively on mpMRI (blinded to pathology results) and 13 were retrospectively identified after correlating with histology. However, no difference in quantitative mpMRI parameters was found for the identified cancer lesions. Representative images of identified and unidentified cancers are shown in Figure 1 and Figure 2. 

Of the identified lesions, 78 lesions (81%) were primarily in the peripheral zone, while 18 lesions (19%) were in the transition zone. A similar zonal distribution was found for the unidentified lesions, with 15 (79%) peripheral zone lesions and 4 (21%) transition zone lesions. The unidentified cancers (1.4 ± 1.0 cm × 0.7 ± 0.4 cm) were significantly smaller (*p* < 0.001) in size compared with the identified cancers (2.1 ± 1.2 cm × 1.0 ± 0.5 cm). The unidentified cancers were predominantly low grade (*p* < 0.001) (11 Gleason 3 + 3, 8 Gleason 3 + 4) and stage (*p* < 0.01) (all 19 were stage T2) compared with the identified cancers (Grade: 20 Gleason 3 + 3, 59 Gleason 3 + 4, 13 Gleason 4 + 3, 4 Gleason 4 + 5; Stage: 68 Stage T2, 27 Stage T3). Additional details can be found in Table 2. 

### 3.2. Quantitative MRI Characteristics

Unidentified cancers had significantly higher ADC (1.34 ± 0.38 vs. 1.02 ± 0.30 μm^2^/ms, *p* < 0.001) and T2 (117.0 ± 31.1 vs. 97.1 ± 25.1 ms, *p* = 0.005) compared with cancers identified on MRI (Figure 3). However, the DCE-MRI parameters from EMM analysis, A or amplitude (110.7 ± 45.3 vs. 129.2 ± 51.1%, *p* = 0.206), α or enhancement rate (5.24 ± 2.84 vs. 6.79 ± 5.08% per s, *p* = 0.263), and β or washout rate (0.031 ± 0.083 vs. 0.081 ± 0.072% per s, *p* = 0.222), were not significantly different between the unidentified and identified cancers. 

### 3.3. Pathologic Characteristics

Nineteen unidentified cancers and twenty-five identified cancers with similar sizes and grades were selected for analysis. While investigating all undetected cancers along with size and grade (Gleason 3 + 3 and 3 + 4) matched identified cancers, the relative density and absolute density of cancer glands were found to be significantly (*p* = 0.04) higher in the cancers identified on MRI compared with the unidentified cancers. A large percentage of the unidentified cancers tended to have lower relative density compared with the surrounding tissue (4/19 = 21% vs. 0/25 = 0%) and to be categorized as sparse cancers (8/19 = 42% vs. 2/25 = 8%) compared with the MR-visible cancers. Detailed results can be found in Table 2. However, the percentage of the Gleason 4 component in Gleason 3 + 4 lesions was only nominally (*p* = 0.15) higher in the identified cancers (20 ± 12%) compared with the unidentified cancers on MRI (15 ± 8%). 

An investigation into the tissue composition of these cancers using matching lesion size and grade showed that nonvisible cancers on MRI had significantly lower epithelium volume, 32.9 ± 21.5 vs. 47.6 ± 13.1% (*p* = 0.034), and higher lumen volume, 20.4 ± 10.0 vs. 13.3 ± 4.1% (*p* = 0.021), compared with identified cancers. However, no difference in stroma volume was found between the unidentified and identified cancers (46.6 ± 19.9 vs. 39.1 ± 13.3%, *p* = 0.154). Representative examples of histologic images and corresponding segmented images for a cancer identified on MRI and a cancer undefined on MRI can be seen in Figure 4. Box plots showing the differences in tissue composition (fractional volumes of stroma, epithelium, and lumen) between identified and unidentified cancers are shown in Figure 5. 

Some additional observations by pathologists, possibly explaining why some cancers were not identified, were: cystic changes adjacent to the cancer lesions (*n* = 2), small cancer focus (*n* = 2), cancers being interspersed between atrophic (*n* = 2) or benign glands (*n* = 4), and dense glands surrounding the cancer (*n* = 1). 

## 4. Discussion

In this study, we investigated why some prostate cancers are not identified on mpMRI by using ground truth reference from whole-mount prostatectomy specimens. Our results showed that unidentified cancers tend to exhibit a lower Gleason score and pathologic stage, smaller size, and lower density of glands compared with the surrounding tissue (sparse cancer lesions), and they have different tissue compositions compared with cancer lesions identified on MRI. Independent from size and Gleason score, tissue composition differences, specifically, a higher proportion of lumen (associated with high T2 and ADC values) and lower epithelium (associated with lower T2 and ADC) in unidentified cancers compared with identified cancers, can explain why some of the prostate cancers cannot be identified on mpMRI.

The unidentified cancers are smaller in size and have predominantly lower Gleason scores. These results are in agreement with previous studies, including the large multicenter PROMIS study [9,10,12,15,16,17,18]. The fact that we see that a higher percentage of the unidentified cancers is non-clinically significant cancer reinforces the role mpMRI plays in risk stratification of men with suspected prostate cancer: detecting clinically significant cancers (Gleason 3 + 4 and above) while missing non-clinically significant cancers (Gleason 3 + 3). Future studies may consider looking at the oncological implications of mpMRI visible versus invisible lesions and whether that may affect decisions on treatment options. A negative correlation between the cancer Gleason score and the ADC [31] and T2 [32] has been noted in the literature. This can also be attributed to tissue composition differences in different Gleason patterns, as shown by a previous study [23], which showed that Gleason pattern 3 is closely similar to normal prostate (preserved glandular morphology), whereas in pattern 4 and higher, there is gradually less lumen and more dense epithelial cells. This leads to a reduction in MR contrast in ADC and T2 maps and T2-weighted images in lower-grade lesions, making detection challenging. 

Differences in quantitative mpMRI parameters were also found. The unidentified cancers had significantly higher ADC and T2 compared with the identified cancers, confirming why these lesions were not visualized on mpMRI. Conversely, the DCE-MRI parameters were not significantly different between the identified and nonidentified cancers. Considering DCE-MRI is used as a secondary sequence to T2W and DWI (primary sequences) for prostate cancer diagnosis, and that up to 50% of prostate lesions are not seen on DCE-MRI [33], it is not surprising that a number of lesions remained invisible on DCE as well. 

We chose identified cancers with sizes and Gleason scores similar to the unidentified cancers for subsequent analysis. While looking at size- and Gleason score-matched cancers, the relative density and absolute density of cancer glands were higher in the identified cancers compared with the unidentified cancers. A large percentage of the unidentified cancers tended to have lower relative density compared with the surrounding tissue, and were categorized as sparse cancers, compared with the identified cancers. This is consistent with a study by Langer et al. [28] that showed that sparse prostate tumors have similar mpMRI (ADC and T2 values) to those of benign peripheral zone tissue, which may limit their detection and the assessment of tumor volume of these sparse cancer lesions in mpMRI. Another study by van Houdt et. al. also reported lower tumor density and heterogeneous tumor morphology in undetected prostate cancers on MRI [20], while Miyai et. al. reported that cancer cells cover a larger area in detected cancers (60.9%) versus undetected cancers (42.7%) [34]. Other histologic differences, such as loose or desmoplastic stroma and the presence of intermixed benign glands, along with cancer glands that represent sparse cancers, have also been reported [19].

We found that the percentage of the Gleason 4 component in the Gleason 3 + 4 lesions was not significantly higher in the identified cancers (20 ± 12%) compared with the unidentified cancers (15 ± 8%). While this result is similar to another study [34], the percentage of Gleason pattern 4 reported was higher for lesions in that study due to a slightly different definition of unidentified cancers and the cohort containing Gleason 4 + 3 unidentified lesions (which, by definition, are expected to have more than 50% Gleason 4 pattern). 

Most importantly, tissue composition differences, specifically, the higher lumen and lower epithelium in unidentified cancers, are likely to be the major contributor to clinically observed variations in different appearances of identified and unidentified cancers on mpMRI and may explain why these lesions were missed on MRI. Most prostate cancers are epithelial in nature and are characterized by rapidly proliferating cancer epithelial cells that concomitantly replace the surrounding stromal cells and luminal space. Epithelial cells are associated with low ADC and T2, while lumen is associated with high ADC and T2. Our results show that identified cancers have similarly higher epithelium and reduced lumen compared with unidentified cancers; therefore, identified cancers have lower ADC and T2, making them easily visible on mpMRI, unlike unidentified cancers. In addition, comparing tissue composition with the previous literature [23,35], the tissue composition of identified cancers in this study (47.6 ± 13.1% epithelium, 13.3 ± 4.1% lumen) matches well with the literature values for Gleason pattern 3 and 4 cancers visualized on MRI and confirmed by pathology (~52–58% epithelium, 11–16% lumen). However, unidentified cancers (32.9 ± 21.5% epithelium, 20.4 ± 10.0% lumen) were found to have tissue compositions closely similar to benign prostatic tissue (~33% epithelium, ~27% lumen), and thus, appear similar to the surrounding benign tissue and remain undetected. Therefore, the biophysical basis of MRI signal changes in prostate tissue can be attributed to tissue composition changes, namely, in differences in the proportions of lumen, epithelium, and stroma. Conventional imaging, measuring ADC and T2 using the mono-exponential signal model, may not be able to isolate signals specific to the cancer cells due to signal averaging when tissue composition changes are not spread out. However, a multicompartment tissue model of the prostate (using the components stroma, epithelium, and lumen) can exploit the distinct biophysical properties of each of these tissue components. This is a basis of some of the newer tissue microstructural imaging models, such as hybrid multidimensional MRI (HM-MRI) [36,37], luminal water imaging [38], VERDICT [39], etc., as well as high b-value non-Gaussian DWI models that have the potential to better identify focal areas of tissue where the tissue composition starts to deviate from the surrounding benign tissue and thus, potentially, to detect lesions that cannot be detected on conventional mpMRI. These results may also suggest the need to develop MRI methods that detect intrinsic properties of cancer cells (i.e., the properties of individual cells) rather than extrinsic properties such as cell density. For example, quadrant analysis of hybrid multidimensional MRI detects signals from enlarged nuclei associated with rapidly growing cancers [40]. Even if cancer cells are dispersed within a voxel, the aggregate signal from these enlarged nuclei may be detectable. However, if cancer cells are dispersed within a voxel, they may not produce a low ADC, but they may have a distinct and detectable signal from cancers cells in quadrant analysis. The use of machine-learning-based or deep-learning-based methods has shown promise in identifying a greater percentage of clinically significant prostate cancers [41,42], and the use of these methods to identify these unidentified lesions may also be considered for future studies. 

There are a few limitations to this study. First, this study is a retrospective single-center study, where MRIs were performed using a single MR vendor with a standardized imaging protocol. Additionally, any changes in hardware or imaging parameters may affect image contrast and thus affect whether a lesion is identified on MRI. Therefore, it would be preferable to validate these results for invisible cancers in both a multivendor and multicenter setting using a large sample size (especially that of MR-invisible lesions), including consideration such as the use of independent measures, such that only one lesion is considered for each patient. Third, because whole-mount prostatectomy specimens were needed for the analysis, there is an intrinsic selection bias in the cohort of MR-invisible cancers, as only subjects referred to surgery were enrolled. These may not reflect the population of men with MR-unidentified prostate cancer who did not undergo surgery. Fourth, there may be small uncertainty in the radiology–pathology correlation. The sectioning of the prostate was performed in approximately the same plane as the MRI images, without the use of any patient-specific sectioning mold. In addition, formalin fixation results in tissue shrinkage, which may affect radiology–pathology registration, but this is not unique to our work. Finally, the labeling of unidentified cancers can be subjective as per the visual interpretation of mpMRI, which has been shown to have large inter-reader variation. Combining the prospectively identified lesions and the lesions retrospectively identified after correlating with histology as identified lesions may introduce a bias. The intention was to have a cohort of MR-unidentifiable lesions that are truly MR-invisible lesions, as some of these retrospectively identified lesions could possibly have been identified by other radiologists. While no differences were found for the identified cancer lesions found prospectively on mpMRI or retrospectively identified after correlating with histology, this could be influenced by background signal in the surrounding benign tissue [43]. 

## 5. Conclusions

Our study reveals that, independent from size and Gleason score, there are tissue composition differences, specifically, a higher lumen (associated with high T2 and ADC) and a lower epithelium (associated with lower T2 and ADC) in unidentified cancers compared with identified cancers, which might explain why some of the prostate cancers cannot be identified on mpMRI. Some of the newer tissue microstructural imaging models, such as hybrid multidimensional MRI [44], luminal water imaging [38], VERDICT [39], etc., or the development of new MRI methods that detect the intrinsic properties of cancer cells, may have the potential to detect these lesions that cannot be detected on conventional mpMRI. 

## Figures and Tables

**Figure 1 cancers-15-05825-f001:**
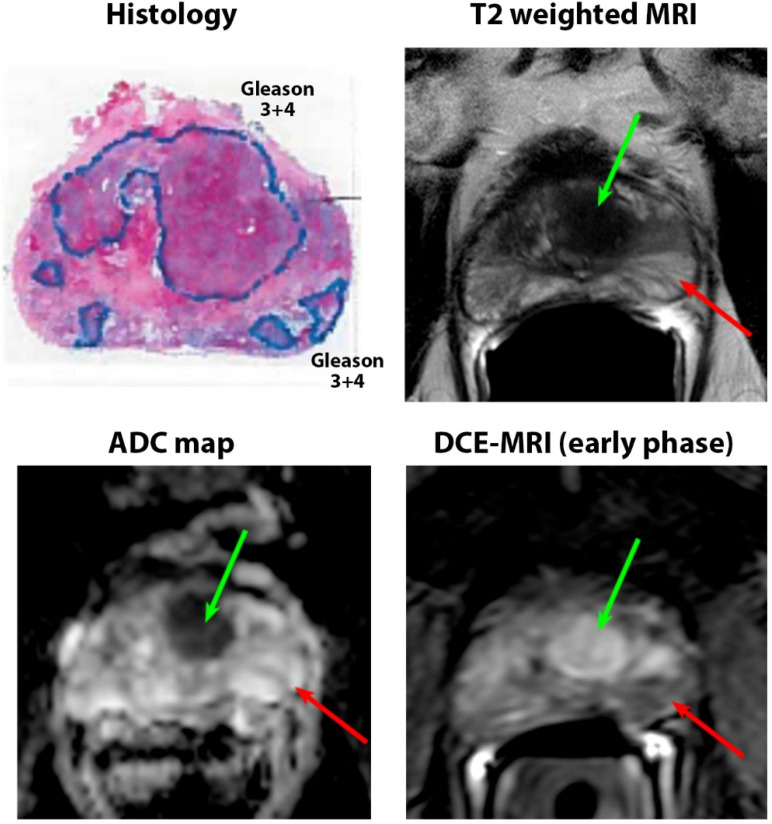
A 67-year-old patient with PSA of 9.5 ng/mL with a Gleason 3 + 4 anterior lesion (green arrow) that was clearly identified on mpMRI. The lesion was highly hypointense on T2W (T2 = 61 ± 8 ms) and ADC (0.58 ± 0.09 μm^2^/ms) and showed focal enhancement on early-phase DCE-MRI. However, another Gleason 3 + 4 lesion in the left peripheral zone (red arrow) was unidentified on MRI. The lesion was isointense on T2W (T2 = 194 ± 34 ms) and ADC (1.51 ± 0.25 μm^2^/ms) and showed no focal enhancement on early-phase DCE-MRI. It was found to have lower glandular density than the surrounding benign tissue and was interspersed between atrophic benign glands.

**Figure 2 cancers-15-05825-f002:**
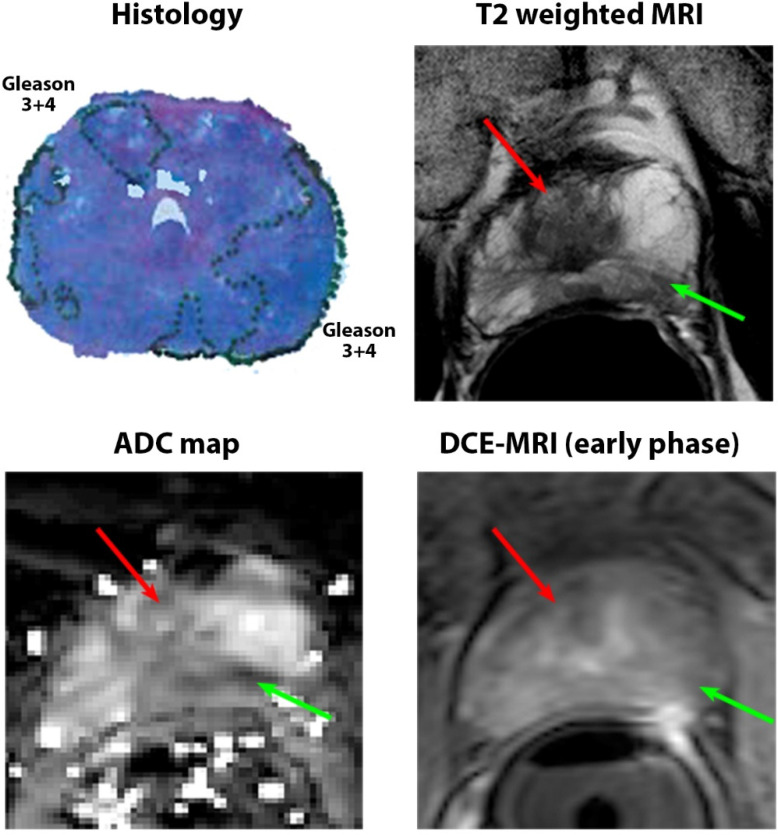
A 53-year-old patient with PSA of 5.1 ng/mL with a Gleason 3 + 4 lesion (green arrow) in the left peripheral zone that was identified on mpMRI. The lesion was focally hypointense on T2W (T2 = 94 ± 34 ms) and ADC (1.13 ± 0.26 μm^2^/ms) but showed no focal enhancement on early-phase DCE-MRI. However, another Gleason 3 + 4 lesion in the right transition zone (red arrows) was unidentified on MRI (no focal area of abnormal MR signal that correlates with cancer presence). The lesion was isointense on T2W (T2 = 138 ± 52 ms) and ADC (1.45 ± 0.41 μm^2^/ms) and showed no focal enhancement on early-phase DCE-MRI, making it indistinguishable from the surrounding benign tissue. Even though it was found to have high glandular density, it was adjacent to very dense benign glands.

**Figure 3 cancers-15-05825-f003:**
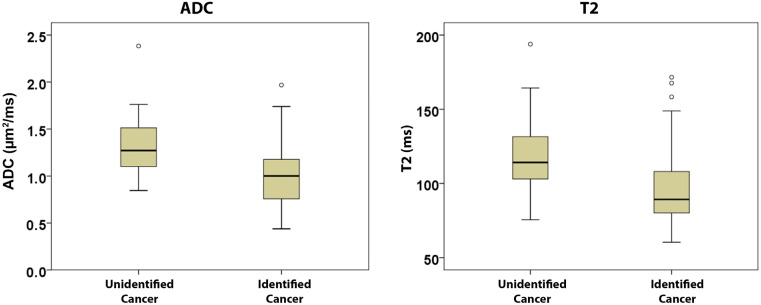
Unidentified cancers had significantly higher ADC (1.34 ± 0.38 vs. 1.02 ± 0.30 μm^2^/ms, *p* < 0.001) and T2 (117.0 ± 31.1 vs. 97.1 ± 25.1 ms, *p* = 0.005) compared with cancers identified on MRI.

**Figure 4 cancers-15-05825-f004:**
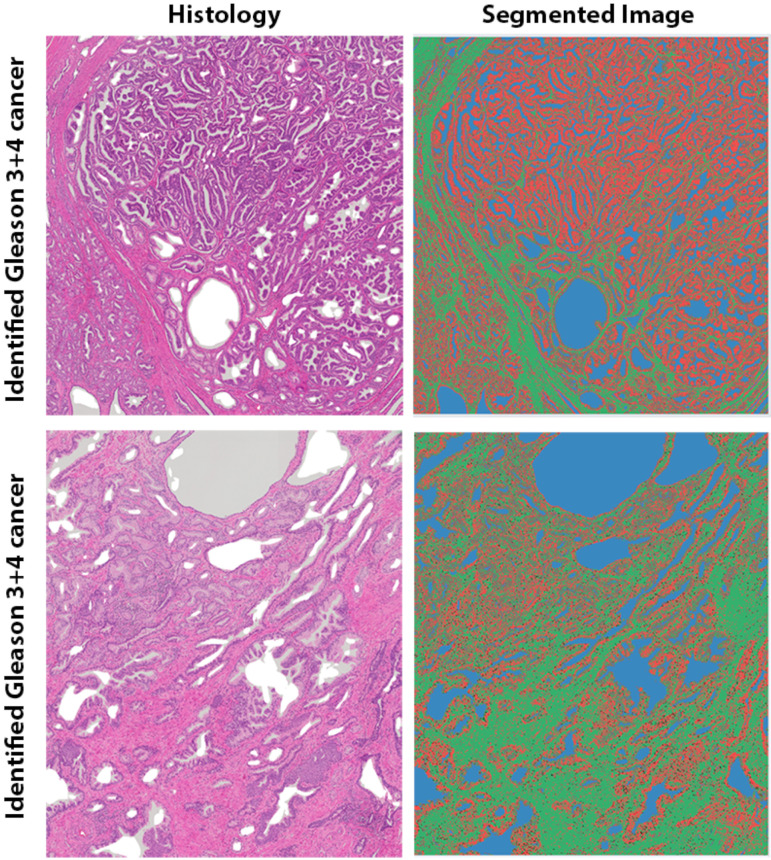
The identified Gleason 3 + 4 cancer from the posterior peripheral zone from the apex midline (**top**) had higher glandular density compared with the MR-unidentified Gleason 3 + 4 cancer from the right peripheral zone (**bottom**) from the same patient (76 years old). Tissue components were segmented into stroma (green), epithelium (red), and lumen (blue). The tissue composition in the identified cancer and the unidentified cancer was different: stroma, 21.5 vs. 45.7%; epithelium, 60.6 vs. 34.0%; and lumen, 17.9 vs. 20.4%. In addition, the identified cancer (40%) had a greater proportion of Gleason pattern 4 than the unidentified cancer (10%).

**Figure 5 cancers-15-05825-f005:**
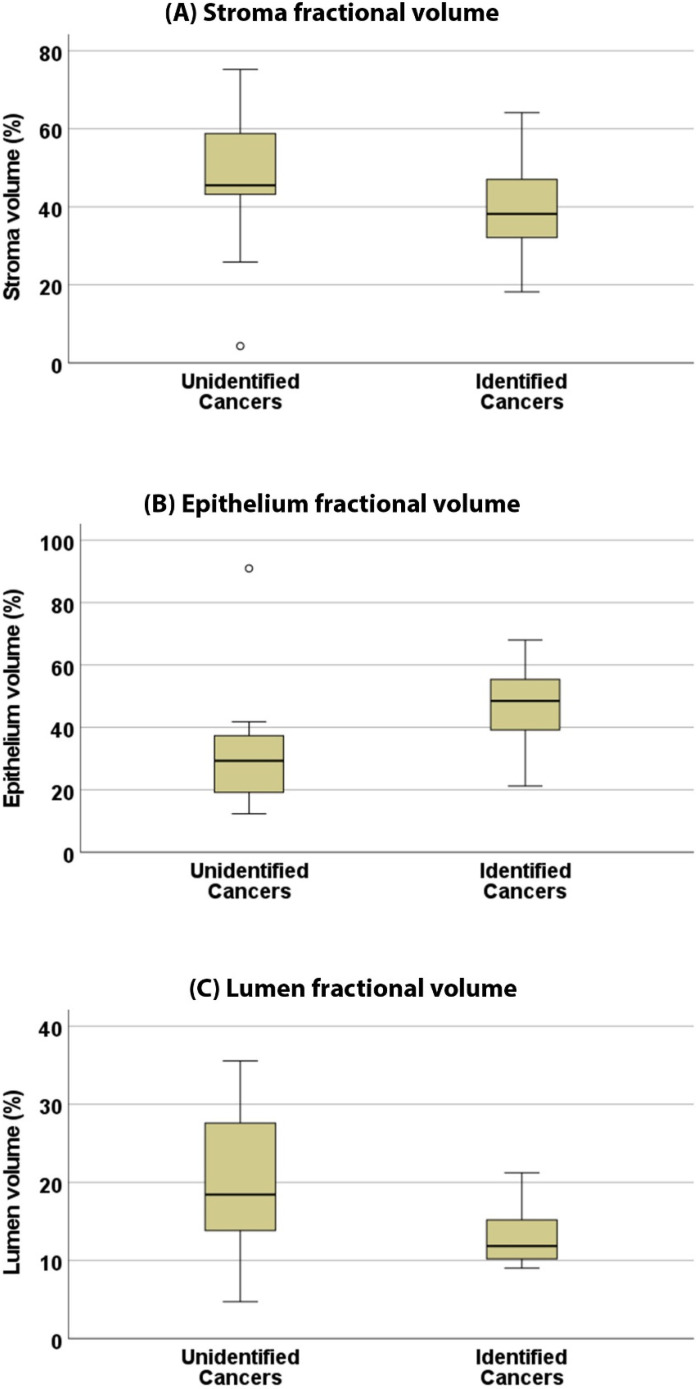
An investigation into the tissue composition of these cancers as presented in these box-plots showed that unidentified cancers had significantly lower epithelium volume (**B**), 32.9 ± 21.5 vs. 47.6 ± 13.1% (*p* = 0.034), and significantly higher lumen volume (**C**), 20.4 ± 10.0 vs. 13.3 ± 4.1% (*p* = 0.021), compared with identified cancers. However, no difference in stroma volume (**A**) was found between the unidentified and identified cancers (46.6 ± 19.9 vs. 39.1 ± 13.3%, *p* = 0.154).

**Table 1 cancers-15-05825-t001:** MR imaging parameters.

Imaging Sequence	Pulse Sequence	FOV (mm)	Scan Matrix Size	In Plane Resolution (mm)	TE (ms)	TR (ms)	Slice Thickness (mm)	Flip Angle (°)
Axial T2W	SE-TSE	160 × 160	400 × 400	0.4 × 0.4	115	8230	3	90
Multi-echo T2W (T2 mapping)	SE-TSE	160 × 160	212 × 212	0.75 × 0.75	30, 60, 90, 120, 150, 180, 210, 240, 270	7850	3	90
DWI ^a^	SE-EPI	180 × 180	120 × 120	1.5 × 1.5	80	6093	3	90
DCE-MRI ^b^	T1-FFE	250 × 385	200 × 308	1.25 × 1.25	3.3	4.8	3.5	10

SE—spin echo; TSE—turbo spin echo; EPI—echo planar imaging; FFE—fast field echo. ^a^ *b*-values used: 0, 50, 150, 990, 1500 s/mm^2^, δ/Δ = ms. ^b^ Contrast agent: gadobenate dimeglumine (MultiHance, Bracco, Minneapolis, MI, USA) was injected at a rate of 2.0 mL/s followed by a 20-mL saline flush. Contrast dose amount was based on patient’s weight (0.1 mmol/kg). DCE-MRI T1-weigthed images were taken with temporal resolution of ~8.3 s at 60 dynamic scan points over 8.2 min.

**Table 2 cancers-15-05825-t002:** Cancer lesion properties.

Gleason Score	Identified Cancers	Unidentified Cancers	All Cancers
3 + 3	20	11	31
3 + 4	59	8	67
4 + 3	13	-	13
4 + 5	4	-	4
Overall	96	19	115
Relative density			
Relative density category	Identified Cancers ^+^	Unidentified Cancers	All Cancers
Significantly less glandular density than surrounding benign tissue (Category 1)	0	1	1
Somewhat less glandular density than surrounding benign tissue (Category 2)	0	3	3
Similar glandular density to surrounding benign tissue (Category 3)	4	3	7
Somewhat higher glandular density than surrounding benign tissue (Category 4)	12	9	21
Significantly higher glandular density than surrounding benign tissue (Category 5)	9	3	12
Overall	25	19	44
Absolute density			
Absolute density category (% of cancer tissue)	Identified Cancers ^+^	Unidentified Cancers	All Cancers
Highly sparse (0–25%) (Category 1)	0	1	1
Sparse (25–50%) (Category 2)	2	7	9
Dense (50–75%) (Category 3)	16	7	23
Highly dense (75–100%) (Category 4)	7	4	11
Overall	25	19	44

^+^ Only size and Gleason score matched identified cancers used in this analysis.

## Data Availability

In accordance with the institutional review board, the data acquired in this study contain person-sensitive information, which can be shared only in the context of scientific collaboration.

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
