# Peer review of "Prostate Cancers Invisible on Multiparametric MRI: Pathologic Features in Correlation with Whole-Mount Prostatectomy"

_cancers, 2023, doi:10.3390/cancers15245825_

Round 1

Reviewer 1 Report

Comments and Suggestions for Authors

The authors present a very interesting an clinically relevant study on prostate cancers invisible on multi-parametric MRI. The used whole mount slides after radical prostatectomy to compare histology and MRI. Most other studies focus on what histology will be revealed in MRI-postitive lesions according e.g. to PIRADS. It is interesting because up to now, it is still not fully understood why some cancers are not visible in MRI. 

One of the findings is that independent from size and Gleason score, tissue composition differences, specifically higher lumen and lower epithelium in unidentifiable cancer lesions (in MRI) can explain why some of the prostate cancers cannot be identified on MRI. 

The number of patients is not very high, but the workload for this investigation seems considerable and acurate. 

English: good

Abstract: good

Manuscript: good

Conclusion : good. 

Figures and tables: good

Author Response

Reviewer #1:

The authors present a very interesting an clinically relevant study on prostate cancers invisible on multi-parametric MRI. The used whole mount slides after radical prostatectomy to compare histology and MRI. Most other studies focus on what histology will be revealed in MRI-postitive lesions according e.g. to PIRADS. It is interesting because up to now, it is still not fully understood why some cancers are not visible in MRI. 

One of the findings is that independent from size and Gleason score, tissue composition differences, specifically higher lumen and lower epithelium in unidentifiable cancer lesions (in MRI) can explain why some of the prostate cancers cannot be identified on MRI. 

The number of patients is not very high, but the workload for this investigation seems considerable and acurate. 

English: good

Abstract: good

Manuscript: good

Conclusion : good. 

Figures and tables: good

Response: Thank you for your kind review and acknowledging that investigation into MR invisible cancers is needed.

We do acknowledge that a study with large sample size is needed, especially with a larger set of MR invisible cancers. This was also noted by reviewer #2 and therefore has been added to the limitations.

Reviewer 2 Report

Comments and Suggestions for Authors

The authors investigated histopathological factors of mpMRI-invisible prostate cancer (PCa) relying on whole-mount sections as diagnostic reference. They found that tissue composition histological differences that are independent of size and grade distinguish invisible and visible PCa at mpMRI.

I have some concerns addressed by the following comments.

-INTRODUCTION:

1) Lines 48-51, mpMRI should not be used as a screening tool as stated by EAU guidelines and it is not an alternative to perform prostate biopsy in the high suspect of PCa. The use of “sample” in this context appears inadequate. Please rephrase.

2) Lines 53-55, the fact that mpMRI misses a large number of tumors is not supported by the literature. This notion is corroborated by the findings of the current study, where only 19 of 119 cancers were invisible. mpMRI yields a high negative predictive value (85%, doi: 10.1097/JU.0000000000000757) for clinically significant PCa and indeed, biopsies can be safely omitted in patients with a low risk of PCa, according to EAU guidelines. Moreover, mpMRI-missed tumors are often clinically insignificant, so omitting biopsy strategies lowers the risk of overdiagnosis and overtreatment.

3) Rather than miss, mpMRI it is known to underestimate the true index lesion pathological volume, especially in small PCa and according to some tumor characteristics (doi: 10.1111/bju.15498.) Please discuss this reference. This may represent the foundation to further investigate invisible vs visible tumors, in a like-to-like fashion.  

4) Line 55-57, it is more the PPV of the mpMRI that varies according to readers and their expertise even though relying on the recent PIRADS scoring systems (doi: 10.1007/s00345-023-04365-4.). Moreover, r reference 11 is incorrect since compares the detection rate of MRI vs TRUS biopsies and not the accuracy of the MRI itself on PCa detection.

-MATERIAL AND METHODS:

1) Line 84-87, unnecessary and it should be removed.

2) There is no mention of the biopsy protocols adopted. How many patients were diagnosed by means of targeted or systematic biopsy or both? How many systematic and targeted biopsies were performed and in which way (cores, route, etc)?

RESULTS

1) How do visible tumors identified only prospectively differ from invisible tumors? The retrospective evaluation of mpMRI scans represents certainly a bias. How many tumors initially invisible were reclassified visible at mpMRI after review?

DISCUSSION

-In the limitation section it should be acknowledged the absolute low numbers of invisible PCa and the heterogeneity coming from retrospective vs prospective evaluation of mpMRI being more likely a potential source of bias.

- Are there some oncological implications for mpMRI visible vs invisible PCa. Do they have the same biological behavior or not? Please discuss

Comments on the Quality of English Language

NA

Author Response

Reviewer #2:

The authors investigated histopathological factors of mpMRI-invisible prostate cancer (PCa) relying on whole-mount sections as diagnostic reference. They found that tissue composition histological differences that are independent of size and grade distinguish invisible and visible PCa at mpMRI.

I have some concerns addressed by the following comments.

 -INTRODUCTION:

1) Lines 48-51, mpMRI should not be used as a screening tool as stated by EAU guidelines and it is not an alternative to perform prostate biopsy in the high suspect of PCa. The use of “sample” in this context appears inadequate. Please rephrase.

Response: This sentence has been rephrased that mpMRI can reliably visualize the whole prostate non-invasively. We do not claim mpMRI is used for screening. I hope you will find this acceptable.

2) Lines 53-55, the fact that mpMRI misses a large number of tumors is not supported by the literature. This notion is corroborated by the findings of the current study, where only 19 of 115 cancers were invisible. mpMRI yields a high negative predictive value (85%, doi: 10.1097/JU.0000000000000757) for clinically significant PCa and indeed, biopsies can be safely omitted in patients with a low risk of PCa, according to EAU guidelines. Moreover, mpMRI-missed tumors are often clinically insignificant, so omitting biopsy strategies lowers the risk of overdiagnosis and overtreatment.

Response: We missed adding the study by Bajgiran et. al. with over 500 prostatectomy cases to back up this statement. This article highlights that using mpMRI around 15% (161/1085) of all lesions are undetected clinically significant cancers on prospective analysis. While ~28% (161/578) of all clinically significantly cancers are not detected. https://link.springer.com/article/10.1007/s00261-018-1823-6

Had we combined invisible lesions with lesion found after retrospective analysis we would have found that 17% of all clinically significant cancers (14/84) or 28% of all cancers (32/115) would have been missed. This is within the range (15-30% clinically significant cares missed) generally reported in the literature. These are significant number of misses.

But the percentage of truly invisible lesions per our definition are less than lesions undetected on prospective analysis as these papers used. However, understanding why these cancers are missed is still important.

3) Rather than miss, mpMRI it is known to underestimate the true index lesion pathological volume, especially in small PCa and according to some tumor characteristics (doi: 10.1111/bju.15498.) Please discuss this reference. This may represent the foundation to further investigate invisible vs visible tumors, in a like-to-like fashion. 

Response: Thanks for pointing this out. We have added the citation along with this statement.

4) Line 55-57, it is more the PPV of the mpMRI that varies according to readers and their expertise even though relying on the recent PIRADS scoring systems (doi: 10.1007/s00345-023-04365-4.). Moreover, r reference 11 is incorrect since compares the detection rate of MRI vs TRUS biopsies and not the accuracy of the MRI itself on PCa detection.

Response: Thank you for pointing out. We acknowledge this and have cited this paper along with one by Westphalen et al. that shows that it also varies between different imaging sites.

-MATERIAL AND METHODS:

1) Line 84-87, unnecessary and it should be removed.

Response: This was added as per journals insistence on adding a statement regarding this. I hope this is acceptable.

2) There is no mention of the biopsy protocols adopted. How many patients were diagnosed by means of targeted or systematic biopsy or both? How many systematic and targeted biopsies were performed and in which way (cores, route, etc)?

Response: All of these patients underwent prostatectomy which was used as the reference standard for our studies as mentioned in section 2.3.

 RESULTS

1) How do visible tumors identified only prospectively differ from invisible tumors? The retrospective evaluation of mpMRI scans represents certainly a bias. How many tumors initially invisible were reclassified visible at mpMRI after review?

 Response: We thank the reviewer for pointing this. Due to the inter reader variability in the interpretation of mpMRI, there could certainly be a bias is what is a MR invisible lesion and visible lesion- either detected prospectively or on retrospective evaluation of mpMRI. We already acknowledged this in limitations (last point).

We have mentioned the number of cancers (n = 13) identified after reassessment - retrospectively identified after correlating with histology. But we found no significant differences between these lesions (ADC 1.16±0.30 um2/ms, T2 115±36ms) based of MRI parameters with lesions found on initial prospective assessment (ADC 0.99±0.30 um2/ms, T2 94±22ms). We decided to club the respectively identified lesions in the identified lesions as these some of these lesions were found to and possibly could have been identified by other radiologists. With this classification we wanted to have a cohort of MR unidentifiable lesions that are a truly MR invisible lesions, rather than having lesions that may have been identified by other radiologists. We have added this in the text as well.

What is considered a lesion found prospectively or after retrospective evaluation with pathology could possibly have been influenced by background signal in the surrounding benign tissue in addition to inter-reader variability, which we have mentioned in the discussion section now.

https://www.ncbi.nlm.nih.gov/pmc/articles/PMC6430663/

DISCUSSION

-In the limitation section it should be acknowledged the absolute low numbers of invisible PCa and the heterogeneity coming from retrospective vs prospective evaluation of mpMRI being more likely a potential source of bias.

Response: We do acknowledge that a study with large sample size is needed, especially with a larger set of MR invisible cancers. This was also noted by reviewer #1 and therefore has been added to the limitations.

- Are there some oncological implications for mpMRI visible vs invisible PCa. Do they have the same biological behavior or not? Please discuss

Response: We don’t not have any data to make a definitive answer for this. Considering most of the invisible cancers are lower grade Gleason 3+3 and 3+4 cancers and smaller in size, we can assume they are likely less aggressive. We have added this as a direction for future studies.

Reviewer 3 Report

Comments and Suggestions for Authors

The study is a retrospective analysis. The topic is interesting. There are some major issues

1) "There is an overlap between some the research subjects used in this study and some of our previously published works [20,21]". The remark is unnecessary as well as the corresponding citations

2)  It is debatable to perform a radiological pathological correlation to complete the assessment of unidentified cancers. Eventually, cancers missed before the pathological examination have been moved from the “unidentified” to the “identified” category. It is far from a “real world” study and may introduce further bias. At least, the number of cancers identified after reassessment should be disclosed

3) Authors did not report any information about the locations of the tumor within the gland (periphery, transitional, anterior zone). I think it would be valuable indeed.

Author Response

Reviewer #3:

The study is a retrospective analysis. The topic is interesting. There are some major issues

1) "There is an overlap between some the research subjects used in this study and some of our previously published works [20,21]". The remark is unnecessary as well as the corresponding citations

Response: This was added as per journals insistence on adding a statement regarding this. I hope this is acceptable.

2)  It is debatable to perform a radiological pathological correlation to complete the assessment of unidentified cancers. Eventually, cancers missed before the pathological examination have been moved from the “unidentified” to the “identified” category. It is far from a “real world” study and may introduce further bias. At least, the number of cancers identified after reassessment should be disclosed

Response: Same response that we gave to reviewer #2.

We thank the reviewer for pointing this. Due to the inter reader variability in the interpretation of mpMRI, there could certainly be a bias is what is a MR invisible lesion and visible lesion- either detected prospectively or on retrospective evaluation of mpMRI. We already acknowledged this in limitations (last point).

We have mentioned the number of cancers (n = 13) identified after reassessment - retrospectively identified after correlating with histology. But we found no significant differences between these lesions (ADC 1.16±0.30 um2/ms, T2 115±36ms) based of MRI parameters with lesions found on initial prospective assessment (ADC 0.99±0.30 um2/ms, T2 94±22ms). We decided to club the respectively identified lesions in the identified lesions as these some of these lesions were found to and possibly could have been identified by other radiologists. With this classification we wanted to have a cohort of MR unidentifiable lesions that are a truly MR invisible lesions, rather than having lesions that may have been identified by other radiologists. We have added this in the text as well.

What is considered a lesion found prospectively or after retrospective evaluation with pathology could possibly have been influenced by background signal in the surrounding benign tissue, which we have mentioned in the discussion section now.

https://www.ncbi.nlm.nih.gov/pmc/articles/PMC6430663/

3) Authors did not report any information about the locations of the tumor within the gland (periphery, transitional, anterior zone). I think it would be valuable indeed.

Response: Thank you for the excellent suggestion. We found no difference in the zonal location of the MR visible and invisible lesions.

Of the identified lesions, 78 lesions (81%) were primarily in the peripheral zone while 18 lesions (19%) were in the transition zone. While similar zonal distribution was found for unidentified lesions, with 15 (79%) peripheral zone lesions and 4 (21%) transition zone lesions.

Round 2

Reviewer 2 Report

Comments and Suggestions for Authors

authors addressed all the concerns

Reviewer 3 Report

Comments and Suggestions for Authors

The paper has been properly amended